# Population Genetic Structures of *Puccinia striiformis* f. sp. *tritici* in the Gansu-Ningxia Region and Hubei Province, China

**DOI:** 10.3390/genes12111712

**Published:** 2021-10-27

**Authors:** Cuicui Wang, Bingbing Jiang, Junmin Liang, Leifu Li, Yilin Gu, Jintang Li, Yong Luo, Zhanhong Ma

**Affiliations:** 1Facility Horticulture Laboratory of Universities in Shandong, Weifang University of Science and Technology, Weifang 262700, China; wangcuicui1111@126.com (C.W.); li_jintang@163.com (J.L.); 2Department of Plant Pathology, China Agricultural University, Beijing 100193, China; judyjiang520@163.com (B.J.); leifu_li@163.com (L.L.); guyilin1987@126.com (Y.G.); ygluo@ucanr.edu (Y.L.); 3State Key Laboratory of Mycology, Institute of Microbiology, Chinese Academy of Sciences, Beijing 100101, China; junminliang0311@163.com

**Keywords:** *Puccinia striiformis* f. sp. *tritici*, population genetics, migration, Hubei, Bayesian assignment

## Abstract

Wheat stripe rust, caused by the fungal pathogen *Puccinia striiformis* f. sp. *tritici* (*Pst*), is a destructive wheat disease in China. The Gansu–Ningxia region (GN) is a key area for pathogen over-summering in China, and northwestern Hubei (HB) is an important region for pathogen over-wintering, serving as a source of inoculum in spring epidemic regions. The spatiotemporal population genetic structure of *Pst* in HB and the pathogen population exchanges between GN and HB are important for estimating the risk of interregional epidemics. Here, 567 isolates from GN and HB were sampled from fall 2016 to spring 2018 and were genotyped using simple sequence repeat markers. The genotypic and genetic diversity of *Pst* subpopulations in HB varied among seasons and locations. Greater genetic diversification levels were found in the spring compared with fall populations using principal coordinate analysis and Bayesian assignments. In total, there were 17 common genotypes among the 208 determined, as shown by a small overlap of genotypes in the principal coordinate analysis and dissimilar Bayesian assignments in both regions, which revealed the limited genotype exchange between the populations of GN and HB.

## 1. Introduction

Wheat stripe rust, caused by the basidiomycete fungus *Puccinia striiformis* Westend. f. sp. *tritici* Erikss. (*Pst*), is a destructive disease worldwide [1,2,3]. China is the largest wheat-producing and wheat-consuming country, and wheat stripe rust annually causes various levels of yield loss [4,5,6]. In 2017, outbreaks occurred in 18 provinces (containing the main wheat-producing areas of China in the northwest, southwest, and east), and the occurrence area reached up to 5.56 million hm^2^ [7].

Understanding the relationships among pathogen population structures from different regions is critical to designing a country-wide disease management strategy. Clearly, the movement of pathogens between over-summering and over-wintering regions is a key issue of interregional epidemics. Such movements promote the completion of the pathogen’s annual cycle on a national scale, allowing it to maintain itself over generations. The inoculum can significantly accumulate in over-wintering regions and serve as the source initiating interregional epidemics [5].

The wheat stripe rust pathogen in China cannot over-summer in most of the eastern wheat-growing regions [5]. The mountainous area in the northwestern region includes Southern and Eastern Gansu, Eastern Qinghai, Northwestern Sichuan, and Southern Ningxia (Figure 1) [8]. The pathogen can over-summer on volunteer seedlings and late-maturing wheat at high elevations. The over-summered populations can migrate to low elevations to infect fall seedlings in the same regions, allowing the pathogen to accumulate during fall and winter. Such an inoculum can also disperse southward and eastward to infect fall seedlings, allowing it to survive winter under favorable conditions [5]. The main regions in which the pathogen can over-winter include the Sichuan Basin, Yunnan, Guizhou, Northwestern Hubei (HB), and Southern Henan [5,8]. In these regions, a large number of urediniospores are propagated continuously in winter owing to relatively warm temperatures, and this confirms the contribution of the over-wintering regions to country-wide spring epidemics. Thus, the quantity of urediniospores may greatly influence the intensity of spring epidemics in the eastern regions, including Shaanxi, Henan, Shandong, and other provinces (Figure 1) [8,9].

Gansu and Ningxia Provinces are the most important pathogen over-summering regions [6]. Wheat is planted from late September in the lowlands to late October at various elevations in the mountains. The pathogen is exchanged frequently among subregions in Gansu and Ningxia Provinces [10]. These two regions may be considered a joint area, providing over-summered inocula for other regions through long-distance dispersal, resulting in interregional epidemics [9].

In addition to the Sichuan Basin, the northwestern part of HB has become another important pathogen over-wintering region in China in recent years. HB Province is adjacent to the southeast of Gansu and close to eastern Sichuan Province (Figure 1). Wheat is planted from late September to mid-October, and stripe rust usually occurs in late November, forming outbreaks in late February in the northwestern area of HB [11]. Northwestern HB, in which *Pst* can over-winter [9], may serve as an important geographic area linking the main pathogen over-summering regions (northwestern China) to the major spring epidemic regions. Additionally, northwestern HB represents the rough northern borderline from which *Pst* can over-winter [8] and accumulate as the initial inoculum of epidemics [12]. This feature may significantly affect the country-wide spring wheat stripe rust epidemics [4]. Many questions regarding the role of the over-wintering areas on interregional epidemics still remain. The population genetic structure of *Pst* in HB is unclear. Previous studies have shown that CYR34 has become the main race in Gansu, Ningxia, Sichuan, Hubei [13], etc., which indicates a certain spread of the pathogen among different regions. It has been proved that the Sichuan Basin could receive the inoculum from Gansu Province in the fall [13]; the genetic relationship between the inocula in the main over-summering regions and HB is still unclear.

Phylogeographical analyses were helpful to study the migration pathway or direction of pathogen dispersal, and molecular markers could greatly improve our ability to infer the dispersal routes of pathogens [3,10]. Regarding molecular markers, although single-nucleotide polymorphism (SNP) has a high level of genetic variation and has been used in studies of many organisms, such as developing SNP markers with the technology of kompetitive allele-specific PCR (KASP), it is still at the initial stage for *Pst* [14]. Simple sequence repeats (SSRs) have been the most commonly used marker in recent years [3,10,15] because of their character of codominant and high polymorphism.

In this study, we compared the diversity levels and genetic structures of *Pst* in the Gansu–Ningxia region (GN) in two fall seasons with those in HB in the subsequent spring seasons to infer the possible migration from GN to HB. The objectives of this study were to (i) determine the genetic structure of *Pst* in different seasons in HB and (ii) identify possible *Pst* migration events between the GN and HB populations using various phylogenetic analysis methods.

## 2. Materials and Methods

### 2.1. Sampling and Reproduction of the Isolates

Samplings were conducted in fall 2016, spring 2017, fall 2017, and spring 2018. The sampling periods were from September to January and from February to May. In fall 2016, diseased wheat leaves showing *Pst* sporulation were collected from fields (Figure 1), covering three districts having six counties in Gansu Province, one district having four counties in Ningxia Province, and three districts having seven counties in HB Province (Table 1). In spring 2017, samples were collected from two districts having three counties in HB. In fall 2017, diseased leaves were sampled from three districts having six counties in Gansu and one district having one county in Ningxia. In spring 2018, samples were collected from three districts having seven counties in HB. In total, 567 isolates were obtained: 249, 96, 113, and 109 collected in fall 2016, spring 2017, fall 2017, and spring 2018, respectively (Table 1).

Because frequent exchanges of *Pst* between Gansu and Ningxia occur, it was reasonable to consider them as one population group [9,10]. Individuals sampled in GN in the fall season of 2016 (16F) or 2017 (17F) were defined as one subpopulation, 16F-GN or 17F-GN, respectively (Table 1). In HB, the samples in the same city (or district) containing Shiyan (SY), Xiangyang (XY), Suizhou (SZ), and Jingmen (JM) in the spring or fall (season were considered as subpopulations 16F-HB-SY, 16F-HB-XY, 16F-HB-SZ, 17S-HB-SY, 17S-HB-XY, 18S-HB-SY, 18S-HB-XY, and 18S-HB-JM. Thus, a total of 10 subpopulations were sampled (Table 1, Figure 1).

The pathogens from the samples were propagated on seedlings of the susceptible wheat cultivar ‘Mingxian 169’. Approximately eight seedlings were used per 1 L pot. After a sampled leaf was moisturized at 4 °C for 12 h, urediniospores were picked up from a single uredium pustule and rubbed into a drop of water on the leaf surface of the seedlings as the inoculant. The inoculated seedlings were incubated at 8–10 °C with dew for 24 h in the dark to promote infection and then transferred to an incubator at a 17 °C 14 h day:14 °C 10 h night photo cycle. After approximately 13 days, urediniospores were harvested from the lesions. Each isolate was propagated one or two times, as described above, to obtain an adequate amount of urediniospores. The urediniospores were transferred into a 0.5 mL Eppendorf tube, dried in desiccators at 4 °C for 3–4 days, and stored at −20°C for DNA extraction.

### 2.2. DNA Extraction and Simple Sequence Repeat (SSR) Genotyping

Genomic DNAs of the *Pst* urediniospores were extracted with the cetyltrimethylammonium bromide (CTAB) method [16]. In total, 567 DNA samples were extracted. UV absorptions at wavelengths of 260 and 280 nm were measured using a Nanodrop 2000 spectrophotometer (Gene Company Limited, Hong Kong, China) to determine the DNA concentration of each sample. The samples were then diluted to 50 ng DNA/μL for amplification. Three isolates representing each subpopulation were selected for the amplification of the internal transcribed spacer (ITS) region of the ribosomal DNA with the primers ITS1 (5′-TCCGTAGGTGAACCTGCGG-3′) and ITS4 (5′-TCCTCCGCTTATTGATATGC-3′) and sequenced. All isolates were identified as *Puccinia striiformis* f. sp. *tritici* based on the ITS sequences and morphology of the pathogen [9]. In total, 12 pairs of published SSR or microsatellite primers were used to genotype each isolate (Table 2). All the SSR primers were labeled with FAM, ROX, TAMRA, or HEX fluorescence at their 5′ ends (Table 2). The PCRs for primers CPS08, CPS13, CPS27, CPS34, and WSR44 consisted of 1 μL of 10× reaction buffer, 0.8 μL of MgCl_2_ (25 mM), 1.0 μL of dNTP (2.5 mM), 0.5 μL of forward primer (10 μM), 0.5 μL of reverse primer (10 μM), 0.1 μL of Taq polymerase (5 U/mL), 1.0 μL of DNA (50 ng/mL), and 5.1 μL of double-distilled water. The amplification conditions were as follows: 94 °C for 4 min, 35 cycles of 94 °C for 30 s, 56 °C for 30 s, and 72 °C for 30 s, followed by 72 °C for 10 min. The PCRs for primers RJO3, RJO20, RJN3, RJN5, RJN6, RJN8, and RJN13 contained 1 μL of 10× reaction buffer, 0.8 μL of MgCl_2_ (25 mM), 1.0 μL of dNTP (2.5 mM), 1.0 μL of forward primer (10 μM), 1.0 μL of reverse primer (10 μM), 0.1 μL of Taq polymerase (5 U/ml), 1.0 μL of DNA (50 ng/mL), and 4.1 μL of double-distilled water. The corresponding amplification conditions were as follows: 94 °C for 4 min and 35 amplification cycles of 94 °C for 30 s, 52 °C for 30 s, and 72 °C for 30 s, followed by 72 °C for 10 min. All SSR amplifications were performed in an authorized thermal cycler (Eppendorf AG). The PCR products were separated using an ABI 3730 xl DNA Analyzer (Applied Biosystems), and DNA Marker GS500 (35–500 bp) was used as an internal standard.

### 2.3. Data Analysis

GeneMarker 2.2 [21] was used to analyze the allelic configuration of each sample. A genotype accumulation curve was generated to determine the minimum number of loci necessary to discriminate between individuals in the total population using the poppr package in R [22]. The population genotypic diversity was calculated as corrected lambda, which concerned the differences in the sample size among subpopulations using the poppr package in R [22]. Gene diversity was assessed using Nei’s unbiased gene diversity [23]. The subsequent calculations were performed on the clone-corrected data (i.e., every different genotype was represented only once in a population) using the GenClone 2.0 program [24], which reduced the effects of over-representation by certain clones owing to the epidemics. Such over-representations may influence genetic analyses. The genetic diversity within a population was assessed using POPGENE version 1.3.1 [25]. A mean Shannon’s information index of each locus was calculated using the Nei method [26], and the standard error was calculated by dividing the standard deviation output produced by POPGENE version 1.3.1 by the square root of 12 [24]. Shared genotypes among subpopulations in the same season or between successive seasons were calculated using the GenClone 2.0 program [24].

A nonparametric principal coordinate analysis (PCoA) based on Nei’s distances between all the pairs of SSR genotypes was generated using GenALEx 6.5 [27,28]. The distribution of the genotypes among the studied populations was plotted in two dimensions. A model-based Bayesian method implemented in STRUCTURE 2.3.4 [29] was used to identify genetic clusters and to evaluate the extent of admixtures among them. A total of 267 isolates that were clone-corrected from 567 isolates were used in the STRUCTURE software-based analysis. A model allowing admixture and independent allele frequencies among populations was used. All of the *Pst* isolates were assigned into *K* clusters ranging from 1 to 10 on the basis of their clone-corrected SSR multi-locus genotypes. For each simulated cluster *K*, 50 independent runs were repeated, with 100,000 iterations in Monte Carlo Markov chain replications and a burn-in period of 20,000. The best *K* value was obtained by STRUCTURE HARVESTER (http://taylor0.biology.ucla.edu/structureHarvester) and evaluated as Δ*K*, as proposed by Evanno et al. [30]. Indi and pop files of 50 runs corresponding to the best *K* were processed using CLUMPP 1.1.2 [31], and then, a bar plot was generated by Distruct 1.1 [32] using the output of CLUMPP 1.1.2 [31]. The bar plot analysis results were graphed as pie charts for each location on the basis of the membership fraction Q (Q > 0.8) [29]. Thus, individuals with Q > 0.8 in a given cluster were assigned to the cluster, whereas individuals with 0.2 < Q < 0.8 were assigned to the admixture cluster. Genetic distances between isolates was calculated using GenALEx 6.5, and the phylogenetic trees of the 267 isolates were reconstructed by an unweighted pair-group method with the arithmetic means (UPGMA) approach using MEGA 7. Linkage disequilibrium (LD) tests were performed using Multilocus version 3.1 [33]. The significance of the LD was assessed by the standardized index of association (rBarD). The rBarD values ranged from 0 to 1, with 0 indicating a random association of alleles at different loci, namely linkage equilibrium, and 1 indicating a complete association of alleles at different loci (complete LD) [33]. A randomization test, executed 1000 times, was carried out using the hypothesis of rBarD = 0. The statistical significance of the observed rBarD value was assessed by comparison with the 1000 rBarD value generated under the null hypothesis.

## 3. Results

### 3.1. Genotypic Diversity

Among the 567 *Pst* isolates, 208 unique genotypes were identified. During the wheat-growing year of 2016-2017, the eMLG of 16F-HB-SY, 16F-HB-XY, and 16F-HB-SZ was much lower (eMLG < 5) than that of 16F-GN (eMLG = 9); however, it increased in the spring of 2017 according to the results of the rarefaction curve (Appendix A, Table 3). In the next wheat-growing year, the level of eMLG among all subpopulations was similar. The same distribution of genotypic diversity was observed in the corrected lambda (Table 3).

The level of genotype exchange could be determined by assessing the shared genotypes among subpopulations. In accordance with the relationship between the two successive seasons in HB, only 4 (g10, g17, g19, and g20) of 69 genotypes (Appendix A) were shared, and g20 was the dominant genotype (Figure 2c). Of 208 genotypes, 10 were shared between the GN and HB subpopulations in the 2016–2017 season (Figure 2a, b) 7 seven were shared in the 2017–2018 season (Figure 2d). Thus, there was a small level of pathogen population exchange between the GN and HB subpopulations, as revealed by the limited number of common genotypes.

### 3.2. Genetic Diversity

Genetic diversity was estimated using the parameter of Nei’s unbiased gene diversity, and the values in subpopulations in 16F-GN and 17F-GN were similar. Nevertheless, the values in 16F-HB were much lower than those in 16F-GN and increased in the spring of 2017. Except for 18S-HB-JM, Nei’s unbiased gene diversities of 18S-HB were higher than those of 17F-GN (Table 3).

### 3.3. Population Subdivision on Spatial and Temporal Scales

PCoA showed that the horizontal and vertical coordinates represented 21.72% and 13.69%, respectively, of the total genetic variance in the 2016–2017 season and 19.54% and 16.31%, respectively, in the 2017–2018 season (Figure 3). The 16F-GN sample was located in all four quartiles, whereas the 16F-HB-SY, 16F-HB-XY, and 16F-HB-SZ samples were mostly located in the second and third quartiles (Figure 3a). This result indicated a certain separation of the HB population from the GN subpopulation in 16F. However, the genotypes in 17S- and 18S-HB were mixed with parts of genotypes in 16F-GN and 17F-GN, respectively (Figure 3a,b), suggesting that a small level of population exchange exists between the fall subpopulations in GN and the spring subpopulations in HB.

The result of the Bayesian assignment analysis carried out using STRUCTURE revealed three genetic clusters among the 267 isolates (Figure 4). Individuals in the GN subpopulation were assigned into the G1, G2, G3, and admixture groups, indicating a certain degree of diversification (Appendix A, Figure 4a). Three subpopulations in HB in 2016 were mainly assigned into the G3 and admixture groups, revealing a large difference compared to the GN subpopulation in the same season (Figure 4c). Although a number of G1 and G2 isolates appeared in 17S-HB, the composition was more similar to that of 16F-HB than that of 16F-GN. Similarly, in 2017–2018, the proportion of G2 isolates was much lower in 18S-HB than in the previous season in GN.

The phylogenetic analysis showed similar results with PCoA and Bayesian assignment analysis. The isolates of 16F-HB were mainly clustered with 17S-HB, and a small number of isolates in 16F-GN had closed genetic relationships with 17S-HB (Figure 5a). Similarly, some isolates of 17F-GN were mixed with 18S-HB (Figure 5b).

### 3.4. LD Test

The levels of LD (rBarD) between subpopulations of wheat in 2016–2017 and 2017–2018 were distinct. None of the rBarD values were significant in subpopulations present in fall 2016 and spring 2017, which indicated the existence of *Pst* recombination in GN and HB. However, all the rBarD values were significant at *p =* 0.01 in subpopulations present in fall 2017 and spring 2018 (Table 3).

## 4. Discussion

The genetic diversity levels and population structures of seasonal subpopulations of *Pst* in HB were investigated for the first time in this study. Owing to the severe disease epidemics that occurred from 2016 to 2017 in China under the influence of high winter temperatures [7,34], a large number of diseased leaves were collected in fall 2016, which was not common. Previous field investigations indicated that there was no possibility for *Pst* to over-summer in northwestern HB owing to the high temperatures and the absence of *Pst* hosts [12,35]. Thus, the primary inoculum in fall seasons most likely migrated from the over-summering regions. Therefore, the occurrence of wheat stripe rust disease in HB relied on external sources of *Pst*.

There were only 17 of 208 shared genotypes, a limited PCoA overlap, and dissimilar Bayesian assignments, which revealed that the *Pst* populations of GN and HB underwent a low level of pathogen exchange. *Pst* migration might occur between GN and HB after over-summering; however, the amount of inoculum in HB received from GN was limited. Therefore, the disease intensity in fall seedlings of GN was not an indicator of a possible risk of wheat stripe rust epidemics in HB in the spring.

Shared genotypes, PCoAs, Bayesian assignments, and phylogenetic analysis help infer the levels of population exchange. Liang et al. [10] found that 40 of 55 *Pst* genotypes are common between Gansu and Ningxia Provinces, indicating that a frequent pathogen exchange occurs between the two regions. Similarly, Wan et al. [16] found that 7 of 72 *Pst* genotypes are shared between Gansu and Xinjiang subpopulations, illustrating that Xinjiang and Gansu are not completely isolated. Similarly, in Liang et al. [15], a large number of shared *Pst* genotypes, large PCoA overlap, and homologous Bayesian assignments confirmed that migration occurs between the fall population in Gansu and the spring population in the Sichuan Basin. In this study, 17 of 208 genotypes were shared between the GN and HB subpopulations; therefore, these two regions are not isolated completely.

The amount of *Pst* in HB in the early spring could influence the occurrence of wheat stripe rust in further eastern regions because of transfer by western winds [5,36,37]. Therefore, it was crucial to clearly define the origin source of *Pst* in HB after over-summering. A previous study illustrated that *Pst* spores can disperse to HB from the northwest (containing GN) through an analysis of the upper air current above 5500 m [38]. However, in this study, apart from the shared genotypes between the subpopulations in HB and GN, there were 44 genotypes in HB in S17 and 30 genotypes in S18 (Appendix A) that were not detected in previous subpopulations in GN. That implied that the main source of *Pst* came from other regions to HB after over-summering, because the pathogen cannot over-summer locally.

Field investigations confirmed that *Pst* cannot over-summer in northwestern HB because of the high temperature and the absence of wheat in the high-altitude area [31]. Therefore, the inoculum in northwestern HB after summer was most likely from the northwest. In addition to Gansu, Ningxia, Shaanxi, Qinghai, Xinjiang, and Tibet also suffer from severe stripe rust disease (Figure 1) [16,39]. However, previous studies have illustrated that there is limited pathogen exchange between Xinjiang, Tibet, and the other regions of China [16,39]. *Pst* found east of Qinghai could over-summer; however, spring epidemics in Qinghai rely on inocula from other regions because *Pst* cannot survive the winter [40]. Thus, the migration from fall to winter inferred in this study might not have originated in Qinghai. This study illustrated that the frequency of pathogen exchange between GN and HB is low. Thus, *Pst* populations in the middle area (such as western Shaanxi, Figure 1) could migrate to northwestern HB after the summer. Still, the possible migration from the southeast to HB could not be ruled out. Although *Pst* could survive year-round in some locations of Yunnan and Guizhou [9], the detailed locations of over-summering and over-wintering areas in Yunnan and Guizhou and the genetic relationships of *Pst* with subpopulations in other epidemic regions are still not clear. Therefore, future studies should focus on the population genetic structure of *Pst* among populations in western Shaanxi, northwestern HB, Yunnan, and Guizhou to determine the main *Pst* inoculum source in northwestern HB after over-summering.

This study revealed the usefulness of phylogenetic approaches in inferring the genetic relationships of wheat stripe rust pathogen among subpopulations. More sampling sites in HB would help reveal distinct variations compared with the GN populations. Future studies of the long-distance dispersal of *Pst* would benefit from next-generation sequencing (e.g., restriction-site-associated DNA sequencing), which detects more genetic differences than microsatellites. Therefore, this technology would improve assignments that infer the relationships among regions having a high level of gene flow [41,42]. These data will be vital to help determine the geographic range and timing of disease management strategies in different regions.

Interregional disease management strategies rely on fully understanding the interactions among geographic pathogen populations and their possible long-distance migration patterns. Such information significantly enhances the designs of country-wide disease management strategies. For instance, the efficient reduction of the inoculum in source regions would significantly reduce the inoculum in the target regions resulting from long-distance pathogen dispersal. In this study, because HB is considered a region in which *Pst* accumulates during the winter, significantly reducing the amount of inoculum produced from this region becomes an important strategy to reduce the risk of spring epidemics in the northern regions to which the exogenous inoculum is transferred.

## 5. Conclusions

In conclusion, we found various levels of *Pst* genetic and genotypic diversity in HB in different seasons. Although several common genotypes existed in GN and HB, the small overlap in the PCoA, and the dissimilar Bayesian assignment, revealed that the *Pst* populations of GN and HB experience a low level of exchange. This study revealed that *Pst* migration might occur between GN and HB after over-summering; however, the amount of inoculum in HB received from GN was limited. We recommend further investigations on the influence of *Pst* sources from Shaanxi and the southwestern regions of China (such as Yunnan and Guizhou) on *Pst* in HB. They may provide additional clues to increase our understanding of the migratory pathways of *Pst*.

## Figures and Tables

**Figure 1 genes-12-01712-f001:**
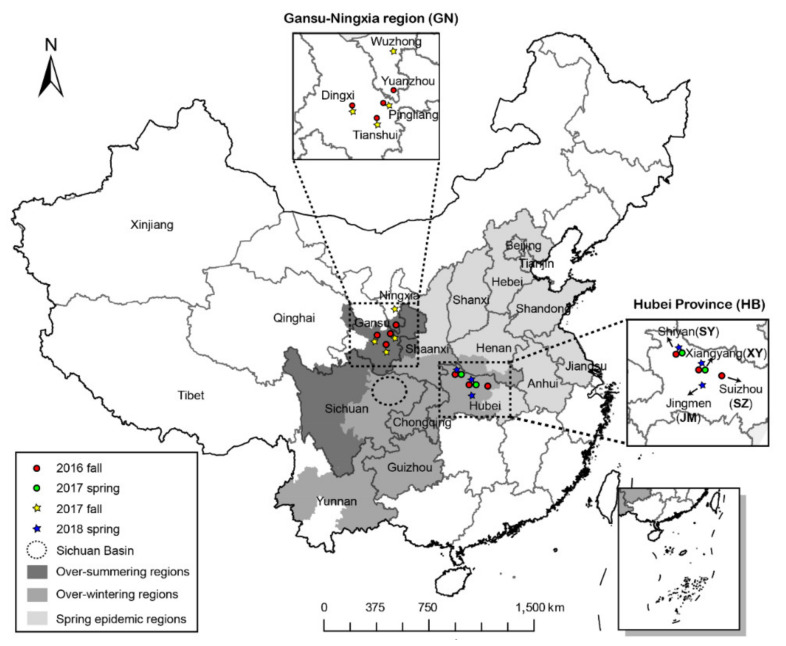
Locations in the Gansu–Ningxia region (5) and Hubei Province (4) from which 567 isolates of *Puccinia striiformis* f. sp. *tritici* were collected in fall 2016, spring 2017, fall 2017, and spring 2018.

**Figure 2 genes-12-01712-f002:**
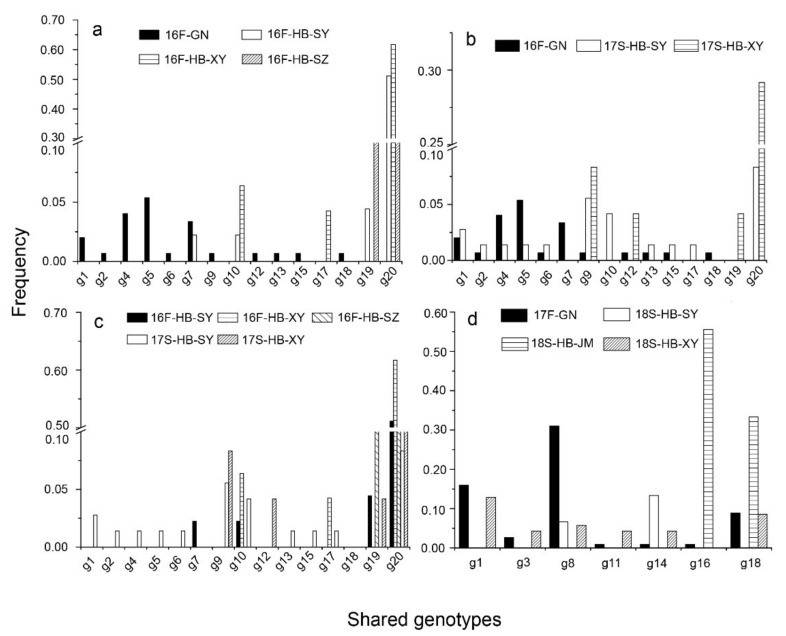
The frequencies of the shared genotypes of *Puccinia striiformis* f. sp. *tritici* among subpopulations of the Gansu–Ningxia region (GN) and Hubei Province (HB) in fall 2016 (**a**), between the subpopulations of GN in fall 2016 and HB in spring 2017 (**b**), between the subpopulations of HB in fall 2016 and spring 2017 (**c**), and between the subpopulations of GN in fall 2017 and HB in spring 2018 (**d**).

**Figure 3 genes-12-01712-f003:**
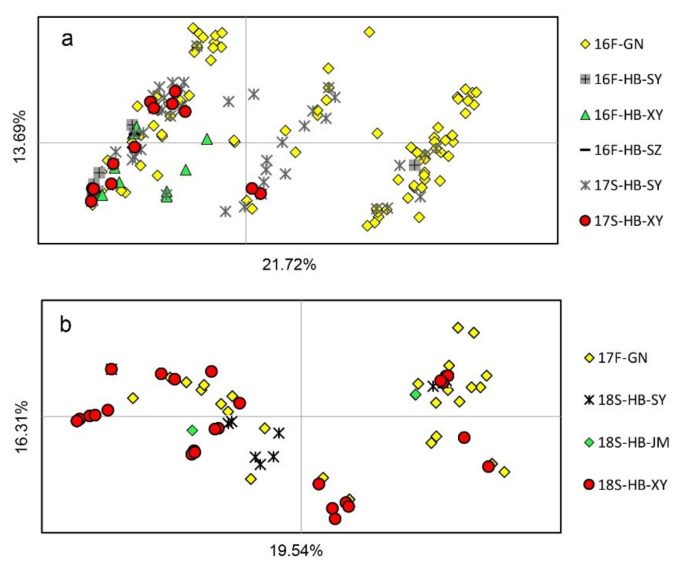
Results of the principal coordinate analysis (PCoA) on seasonal subpopulations of *Puccinia striiformis* f. sp. *tritici* sampled from the Gansu–Ningxia region (GN) and Hubei Province (HB) during fall 2016 (16F) and spring 2018 (18S). Comparisons were carried out among the subpopulations of 16F-GN, 16F-HB, and 17S-HB (**a**) and between the 17F-GN subpopulation and the 18S-HB subpopulation (**b**).

**Figure 4 genes-12-01712-f004:**
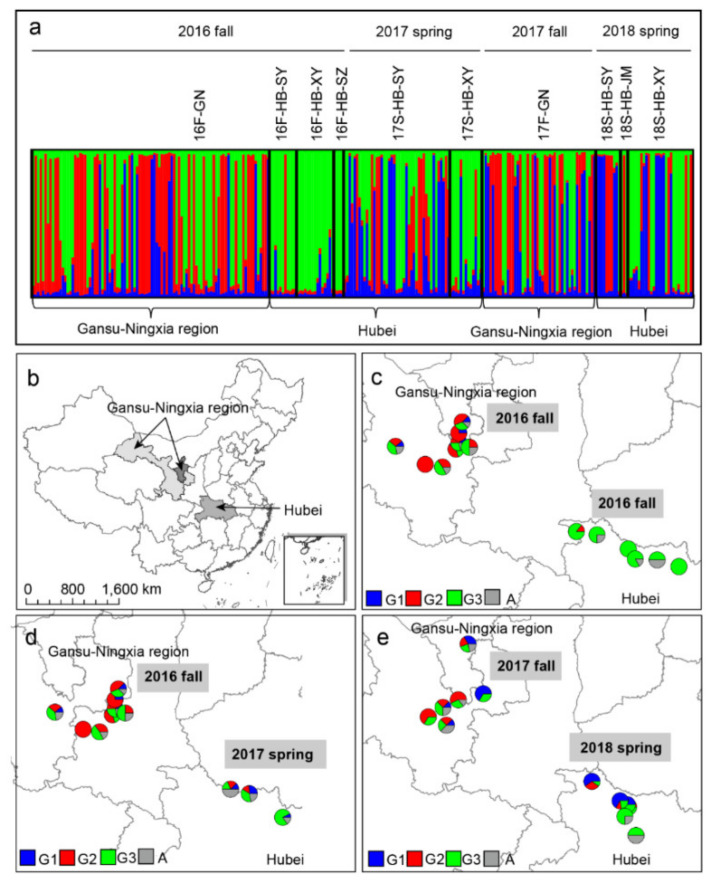
Assignment of isolates to microsatellite genotypes and the geographical distribution of the different genotypic groups of *Puccinia striiformis* f. sp. *tritici* among the sampling sites, as determined using STRUCTURE software (**a**). The geographical locations of samples collected in the Gansu–Ningxia region (GN) and Hubei Province (HB) (**b**). Geographical distributions of genetic group (G) memberships, showing the proportions of individuals assigned to the G1 (blue), G2 (red), G3 (green), and admixture (light gray) groups at each site in 16F-GN vs. 16F-HB (**c**), 16F-GN vs. 17S-HB (**d**), and 17F-GN vs. 18S-HB (**e**), as determined using STRUCTURE. Each vertical line represents an individual having its genome partitioned into *K* segments (here, *K* = 3 genetic groups, shown in blue, red, and green).

**Figure 5 genes-12-01712-f005:**
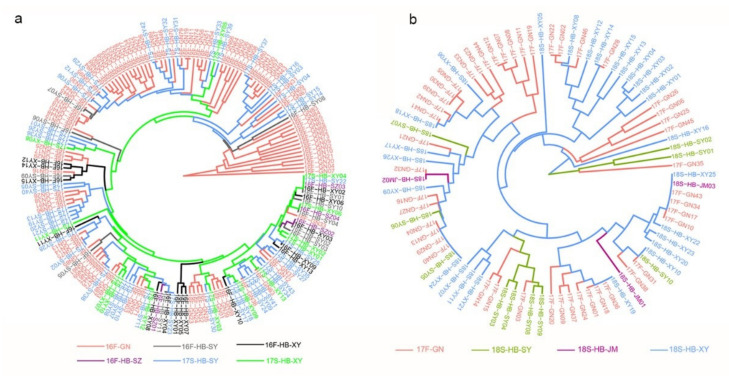
UPGMA tree showing the phylogenetic relatedness among the 267 *Pst* isolates sampled from the Gansu–Ningxia region (GN) and Hubei Province (HB) during fall 2016 (16F) and spring 2018 (18S). Comparisons of genetic relationship were carried out among subpopulations of 16F-GN, 16F-HB, and 17S-HB (**a**), and between the 17F-GN subpopulation and the 18S-HB subpopulation (**b**). Nei’s genetic distance was calculated using GenALEx 6.5, and the phylogenetic tree was reconstructed in Mega 7.

**Table 1 genes-12-01712-t001:** Information on *Puccinia striiformis* f. sp. *tritici* samples collected from the Gansu–Ningxia region and Hubei Province of China from fall 2016 to spring 2018 in this study.

Subpopulation ^a^	Size_sub_ ^b^	Sampling Time (Date/Month)	City	County	Size_cou_ ^c^	Elevation (m)	Cultivar
16F-GN	149	23/11	Tianshui	Gangu	14	1810	Unknown
		22/11		Qingshui	1	1560	Unknown
		23/11		Qinzhou	3	1710	Unknown
		24/11	Dingxi	Longxi	4	2090–2120	Unknown
		23/11		Lintao	20	Unknown	Unknown
		22/11	Pingliang	Zhuanglang	11	1640	Unknown
		23/11	Guyuan	Longde	37	1479–2082	Zhuang 88, Zhongyin 16, Laoxuan 1,
		21/11		Jingyuan	20	1704–1919	Zhongyin 3, Longjian 127, Qinmai 4, Ningdong 3, Ningdong 5
		23/11		Pengyang	14	1479–1633	Unknown
		23/11		Yuanzhou	25	1595–1832	Ningdong 1, Longzhong 6
16F-HB-SY	45	22/12	Shiyan	Yunyang	15	190	Unknown
		23/12		Yunxi	30	170–400	Mianyang 26
16F-HB-XY	47	21/1	Xiangyang	Laohekou	20	Unknown	Unknown
		19/1		Xiangcheng	20	Unknown	Unknown
		17/1		Zaoyang	7	94	Unknown
16F-HB-SZ	8	17/1	Suizhou	Suizhou	8	Unknown	Unknown
17S-HB-SY	72	29/4	Shiyan	Yunyang	44	205-680	Echun series
		1/5		Yunxi	28	Unknown	Mianyang 31
17S-HB-XY	24	20/4	Xiangyang	Fancheng	24	Unknown	Unknown
17F-GN	113	23/11	Tianshui	Qinzhou	4	1630–1690	Zhongliang
		23/11		Maiji	15	1700	Zhongliang
		24/11	Dingxi	Longxi	13	2190	Unknown
		24/11		Tongwei	30	1850–1990	Lantian
		23/11	Pingliang	Kongtong	9	1490	Pingliang 42–44
		22/11		Zhuanglang	19	1650–1740	1028, 1038, Lantian 26
		16/11	Wuzhong	Tongxin	23	Unknown	Unknown
18S-HB-SY	30	23/3	Shiyan	Yunyang	30	167–374	Xiangmai75, Emai596, Zhengmai005
18S-HB-JM	9	19/3	Jingmen	Zhongxiang	9	40	Huayu911,
18S-HB-XY	70	2/4	Xiangyang	Nanzhang	14	Unknown	Unknown
		29/3		Yicheng	11	Unknown	Xiangmai55, Zhengmai9023
		1/4		Laohekou	24	Unknown	Unknown
		1/4		Fancheng	11	Unknown	Unknown
		1/4		Xiangcheng	10	70	Zhengmai101, Zhouheimai1
Total	567						

^a^ GN and HB represent the Gansu–Ningxia region and Hubei Province, respectively, and 16F, 17S, 17F, and 18S represent the fall of 2016, spring of 2017, fall of 2017, and spring of 2018, respectively. ^b^ Size_sub_ indicates the number of isolates in each subpopulation. ^c^ Size_cou_ indicates the number of isolates from each county.

**Table 2 genes-12-01712-t002:** Information on the SSR primers for *Puccinia striiformis* f. sp. *tritici* used in this study.

Locus	Repeat Motif	Primer Sequence (5′–3′)	T_a_ (°C)	No. of Alleles (Size)	Reference
CPS8	(CAG)_14_	F: FAM-GATAAGAAACAAGGGACAGC R: CAGTGAACCCAATTACTCAG	55	5 (200–212)	[17]
CPS13	(GAC)_6_	F: FAM-TCCAGGCAGTAAATCAGACGC R: ATCAGCAGGTGTAGCCCCATC	58	2 (125–128)	[17]
CPS27	(TTC)_4_	F: TAMRA-GATGGGGAAAAGTAAGAAGT R: GGTGGGGGATGTAAGTATGTA	57	2 (225–228)	[17]
CPS34	(TC)_9_	F: TAMRA-GTTGGCTACGAGTGGTCATC R: TAACACTACAAAAGGGGTC	55	5 (104–114)	[17]
RJO3	(TGG)_8_	F: FAM- GCAGCACTGGCAGGTGG R: GATGAATCAGGATGGCTCC	52	4 (201–212)	[18]
RJO20	(CAG)_4_	F: HEX-AGAAGATCGACGCACCCG R: CCTCCGATTGGCTTAGGC	52	3 (283–289)	[18]
RJ3N	(CT)_9_	F: ROX-TGGTGGTGCTCCTCTAGTC R: AGGGGTCTTGTAAGATGCTC	52	4 (335–343)	[19]
RJ5N	(CT)_8_	F: ROX-AACGGTCAACAGCACTCAC R: AGTTGGTCGCGTTTTGCTC	52	3 (223–229)	[19]
RJ6N	(AAC)_9_	F: TAMRA-CAATCTGGCGGACAGCAAC R: CACCTAGGATACCACCGCC	52	4 (309–318)	[19]
RJ8N	(GAT)_8_	F: FAM-ACTGGGCAGACTGGTCAAC R: TCGTTTCCCTCCAGATGGC	52	6 (301–330)	[19]
RJ13N	(ACG)_6_	F: HEX-TTAGCTCAGCCGGTTCCTC R: CAGGTGTAGCCCCATCTCC	52	2 (149–152)	[19]
WSR44	(GT)_6_	F: HEX-AGGCCCCAGGAACACAAAAA R: TCACACACGCTCCACAGTAC	56	2 (188–190)	[20]

**Table 3 genes-12-01712-t003:** Analyses of genotypic and genetic diversity levels of the Gansu–Ningxia and Hubei subpopulations of *Puccinia striiformis* f. sp. *tritici* sampled from fall 2016 to spring 2018.

Subpopulation ^a^	*n*	MLG ^b^	eMLG ^c^	Lambda	Corrected Lambda ^d^	No. of Clones	^e^ Standardized Index of Association (rBarD)	Nei’s Unbiased Gene Diversity
16F-GN	149	90	9.01	0.97	0.97	96	0.0065	0.27
17F-GN	113	34	6.59	0.86	0.87	46	0.0736 **	0.29
16F-HB-SY	45	11	4.44	0.68	0.69	11	−0.034	0.16
17S-HB-SY	72	43	9.00	0.96	0.98	43	0.055	0.25
18S-HB-SY	30	10	6.03	0.84	0.87	10	0.1765 **	0.30
16F-HB-XY	47	15	4.63	0.61	0.62	15	0.040	0.17
17S-HB-XY	24	13	7.15	0.86	0.90	13	0.024	0.19
18S-HB-XY	70	26	7.47	0.91	0.93	26	0.0579 **	0.31
16F-HB-SZ	8	4	4	0.66	0.75	4	−0.115	0.18
18S-HB-JM	9	3	3	0.57	0.64	3	NA	0.26
Total	567	208	8.79	0.96	0.97	267	0.2320 **	0.28

^a^ GN and HB represent the Gansu–Ningxia region and Hubei Province, respectively, and 16F, 17S, 17F, and 18S represented the fall of 2016, spring of 2017, fall of 2017, and spring of 2018, respectively. ^b^ MLG represents the number of multilocus genotypes or genotypic richness observed. ^c^ eMLG represents the number of expected MLGs based on the rarefaction curve. ^d^ Corrected lambda, a parameter of genotypic diversity, was calculated as lambda *(*N*/*N-1*). ^e^ Multilocus linkage disequilibrium (LD) was assessed using the standardized index of association (rBarD) and estimated for each population, as described previously [33]. The significance of rBarD was tested with 1000 randomizations of the data by comparing the observed value to the expected value under the null hypothesis of rBarD = 0. The null hypothesis of LD was rejected if *p <* 0.01. ** Significant at *p =* 0.01.

## Data Availability

Not applicable.

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
