# Peer review of "Population Genetic Structures of Puccinia striiformis f. sp. tritici in the Gansu-Ningxia Region and Hubei Province, China"

_genes, 2021, doi:10.3390/genes12111712_

Round 1
Reviewer 1 Report
Here are some of my comments:
1. Having 208 genotypes for 567 samples is too high diversity that means on average there are about 2-3 isolates for each genotype. I believe the number of samples is too low when you report such a high diversity. My suggestions is to focus on main genotypes perhaps the authors drop some of the markers to make the genotypes much less.
Also with such high diversity the results will not be useful for any disease management plan, contrary to what the authors say in the introduction.
2. Table 3 should be arranged based on regions such as put all GN one after the other so that the reader can see the seasonal variation of genotypes better.
3. Present the shared genotypes Fig 2 differently. The way presented is not clear.
4. Identity of the collected samples at least for some should be identified by ITS sequence and be included in the material and methods.
5. The relationship of genotypes is not presented. Is it possible to do that as a phylogenetic tree, for example?
Reviewer 2 Report
Diversity indices analysis needs to be revised as the calculations didn't seem to take into account the different population sizes, which would affect the diversity indices used (example: genotypic diversity and Shanno's) to be able to establish a comparison among populations and seasons.
In the methodology for a single isolate increase, how was potential contamination determined? did virulence tests were determined on a set of differentia lines to determine purity? is there a way to determine potential contamination via your genotyping analysis? more than one genotype in. a sample can result in erroneous genotyping and diversity assessment.
Reviewer 3 Report
The manuscript of author Cuicui Wang et. al. (genes-1314731-peer-review-v1) shows the genotypic and genetic diversity of Pst subpopulations in Hubei varied among seasons and different locations. Based on SSR marker analysis 17 common genotypes among the 208 determined, found a small overlap of genotypes in the principal coordinate analysis and dissimilar Bayesian assignments in both regions. That revealed the limited genotype exchange between the populations of the GN and HB region of China.
Authors have done as usual analysis mostly conducted by most of the researchers in genetic diversity and population genetic study. However, here the major concern is population genetic studies require high numbers of markers to draw reliable inferences regarding the genetic relatedness of different population groups or subgroups and provide better precision. Can you please explain this?
I know some studies that showed very limited SSR markers for genetic diversity and population genetics but recent advancement and low-cost availability of NGS tools still conducting genetic diversity study with such a low number of SSR markers is unusual how the author would explain this? Apart from this
I have some comments and suggestions for the authors.
In total, there were 17 common genotypes among the 208 determined, as shown by a small overlap of genotypes in the principal coordinate analysis and dissimilar Bayesian assignments in both regions, which revealed the limited genotype exchange between the populations of the Gansu-Ningxia Region and Hubei Province, China
Abstract
Please write Gansu-Ningxia Region (GN) as repeated afterward in abstract similarly Hubei (HB) and use GN and HB thereafter in the abstract.
Introduction
Page 1, line 39
A grammatical error in a sentence so please write regions “are” instate of “is” and “design” as “designing”
So correct sentence as “Understanding the relationships among pathogen population structures from different regions are critical to designing a countrywide disease management strategy”
Secondly, in the introduction, nothing is mentioned about what type of molecular markers have been used by the different researchers to assess genetic diversity in Puccinia striiformis f. sp. tritici (Pst) or similar pathogen and significance of markers. Also include what type of wheat (common, durum or triticale or any other, etc.) is grown in mentioned regions and what types of isotypes previously observed from the regions.
Hence, I would like to suggest it would be more impactful if the author modify the introduction section with more information.
Materials and Methods
Author, not mentioned the isolates these isolates are virulent or avirulent type please explain?
In addition, Table 1 is not consistent or proper year /season and province mentioned, Secondly it can be modified as No need for column “year, season” and “province “Just mentioned in a footnote what is 16F/R, 17F/R, and 18F/R similarly HB, NG for the province”. In addition, the Table 3 Season column can be deleted and mentioned in a footnote similarly as mentioned above.
Page 6, Line 156- written 13 d and following it, written Days please write consistent d or Days it’s not clear what is d means at a first incident.
Please write complete CTAB name, and ddH2O write as (double distilled water) in the bracket.
Figure 2. Heading and subheadings are not appropriate and confusing please check and revise them.
Results,
As the author mentioned in the introduction “Identify possible Pst migration events between the GN and HB populations using various phylogenetic analysis methods,” I would suggest performing and represent phylogenetic analysis in the results.
Round 2
Reviewer 2 Report
The concern with the indices is not the size of samples, the concern is that the indices utilized need to take into account the different sizes of the samples. In other words, as calculated, your results are not valid. Please look at rarefaction curves and normalizing indices by sample size before comparing population diversity based on different sizes of samples across populations as the results and your conclusions are not correct in their current form.
About the purity of the isolates. On one sampled leave, there could be more than one type of isolate or genotype. if a single uredinium was not carefully isolated and maintain more than one genotyped could have been genotyped. How was the purity of sample/ no-cross contamination determined?
Reviewer 3 Report
NA I am satisfied with the correction and response of the Authors.
Author Response
Thank you very much for your affirmation and approval to our manuscript.